# Impulsivity as Early Emerging Vulnerability Factor—Prediction of ADHD by a Preschool Neuropsychological Measure

**DOI:** 10.3390/brainsci11010060

**Published:** 2021-01-06

**Authors:** Ursula Pauli-Pott, Katja Becker

**Affiliations:** 1Department of Child and Adolescent Psychiatry, Psychosomatics and Psychotherapy, Philipps-University of Marburg, Hans Sachs Str. 6, D-35039 Marburg, Germany; Katja.Becker@med.uni-marburg.de; 2Center for Mind, Brain and Behavior (CMBB), University of Marburg and Justus Liebig University Giessen, Hans-Meerwein-Straße 6, D-35032 Marburg, Germany

**Keywords:** impulsivity, cognitive control, inhibitory control, executive functions, ADHD, ODD, externalizing disorders

## Abstract

Impulsivity, comprising deviations of brain-based bottom-up and top-down control processes, has been regarded as a crucial, early emerging marker of a developmental pathway to attention-deficit/hyperactivity (ADHD) and externalizing disorders. In two independent studies (a cross-sectional study and a longitudinal study), we analyzed the concurrent and predictive validity of a task-based neuropsychological impulsivity measure for preschool children. The sample of Study 1 comprised 102 3–5-year-old children (46% boys). In Study 2, 138 children (59% boys) with elevated ADHD symptoms were recruited and assessed at the ages of 4–5 and 8 years. In both studies, preschool impulsivity was measured by a summary score of neuropsychological tasks on approach motivation and hot inhibitory control. For Study 1, the impulsivity measure was significantly associated with symptoms of ADHD and oppositional defiant disorder (ODD) (χ^2^(1) = 9.8, *p* = 0.002; χ^2^(1) = 8.1, *p* = 0.004). In Study 2, the impulsivity measure predicted the 8-year-olds’ ADHD diagnoses over and above concurrent ADHD symptoms (χ^2^(1) = 10.0, *p* = 0.002, OR = 5.0, 95% CI: 1.8–14.0). The impulsivity measure showed good concurrent and predictive validity. The measure can be useful for the early identification of children at risk for developing ADHD and externalizing disorders.

## 1. Introduction

Several developmentally oriented etiological models on attention-deficit/hyperactivity disorder (ADHD) and externalizing disorders assume that impulsivity constitutes a core deficit which appears early in development. The trait impulsivity model, for example, postulates that impulsivity is a crucial vulnerability factor associated with the development of ADHD and subsequent externalizing disorders, such as oppositional defiant disorder (ODD), conduct disorder (CD), and substance use disorder [1,2]. It is assumed that a primary dysfunction of the brain reward system (bottom-up processes) leads to an excessive behavior approach. Combined with poor top-down control ability, and impulsive behavior, ADHD and externalizing disorders might develop [3]. Etiological models of ADHD postulated multiple causal pathways. One of these pathways is assumed to be characterized by a primary dysfunction of the mesolimbic reward system, leading to the development of hyperactivity or impulsivity symptoms in preschool years and in the course of development toward symptoms of ODD and inattention [2,4,5,6,7,8,9].

Impulsivity is a complex construct which can be defined as the tendency to show stimulus-provoked automatic behaviors and non-reflective approach reactions in reward-related contexts, based on a preference for smaller immediate rewards over larger delayed rewards [10]. This characteristic is thought to be based on two independent components: deviations of bottom-up processes (dysfunctions of the brain reward system) and deficits in top-down control (dysfunction or maturational delay of the frontal brain circuitry). Impulsivity thus overlaps with the neuropsychological constructs of delay aversion [4], hot reward-related inhibitory control [9,11], and high approach motivation [7,12]. These neuropsychological characteristics have been regarded as basal, early emerging markers of the motivational pathway to ADHD and ODD.

In recent years, many studies have analyzed the associations of ADHD and ODD and CD (diagnoses and symptoms) with performance in neuropsychological tasks and reward-related functions. With regard to ADHD, meta-analyses of these studies found medium-sized differences between children with ADHD and healthy comparison groups in tasks capturing the preference for an immediate reward (e.g., monetary incentive delay and temporal reward discounting tasks) in middle and late childhood and adolescence [13] and poor hot inhibitory control (IC) (i.e., low ability to wait for gratification in the preschool period) [14]. In samples of approximately 3–5-year-old children, the overall mean effect size for hot IC tasks (which require suppressing the approach of a rewarding stimulus, according to a rule) was of a large magnitude and significantly larger than in older samples. Moreover, the mean effect size for these tasks exceeded the effect sizes for other putatively ADHD-related executive and attentional functions [14]. This pattern might have had several causes. One possibility is that the reward-related basic deficits of ADHD and ODD can most validly be captured in the preschool period.

Given the centrality of early emerging impulsivity symptoms in etiological models in ADHD and ODD (the assumption that these symptoms constitute an early vulnerability marker) and the seemingly good validity of neuropsychological measures on hot IC and behavioral approach tendencies in preschool years, it seems worthwhile to analyze whether these measures allow for the early identification of children at risk of developing ADHD and ODD. Longitudinal research on the issue, however, is scarce. Marakovitz and Campbell [15] followed a sample of hard-to-manage preschool boys and found that a resistance-to-temptation task at the age of 4 years was associated with symptoms of ADHD at the age of 9 years. Incremental predictive effects (i.e., whether the task predicted the 9-year-old ADHD symptoms over and above concurrent 4-year-old ADHD symptoms) were not assessed. Campbell and von Stauffenberg [16] analyzed a subgroup of the NICHD Study of Early Child Care and Youth Development. A forbidden toy task (at 3 years old) and a delay-of-gratification task (at 5 years old) significantly predicted ADHD symptoms at the age of 9 years, over and above parental education level, the gender of the child, and preschool externalizing symptoms. Breaux, Griffith, and Harvey [17] examined children with prominent ADHD symptoms and found that a gift delay task at the age of 3 years significantly predicted an ADHD diagnosis at the age of 6 years. However, the task did not add significant incremental predictive validity over and above ADHD symptoms at 3 years.

Sample characteristics (general population vs. high-risk sample), the measurement of ADHD outcomes (symptoms vs. diagnosis), and the psychometric properties of the neuropsychological tasks might have contributed to the different findings. Moreover, exclusively hot IC tasks were used. It could be worthwhile to add tasks concerning behavioral approach tendency. A measure which validly captures early task-based impulsivity would be of high utility. Such a measure would not only allow analyses of developmental trajectories to ADHD and ODD but, moreover, could improve the early identification of children at risk. Therefore, we aimed to analyze the concurrent and predictive validity of a neuropsychological measure on hot IC and approach tendency (i.e., a task-based impulsivity measure) in preschool ages (1) in a cross-sectional study involving a community-based sample of preschool children, where the concurrent validity of the task-based impulsivity measure was analyzed, and (2) in a longitudinal study involving a high-risk sample of children with elevated ADHD symptoms, where the predictive validity of the impulsivity score was analyzed. The following hypotheses are tested:Study 1 (concurrent validity): The impulsivity score discriminates between children showing high vs. low ADHD and ODD symptoms;Study 2 (predictive validity): In a sample of preschool children at risk for ADHD, the impulsivity score predicts an ADHD diagnosis at school age and adds incremental predictive validity over and above concurrent (preschool) parent- and teacher-reported ADHD symptoms.

To further explore the validity of the impulsivity score, we aim to identify an optimal cut-off for the discrimination between children with and without high symptoms of ADHD and ODD in Study 1. In Study 2, this cut-off score is used to predict the school age ADHD diagnosis.

## 2. Study 1: Materials and Methods

### 2.1. Participants and Procedure

The sample comprised 102 parent–child pairs recruited from child care centers. Children were aged 3–5 years (m = 54.6, s = 11.5 months). The following exclusion criteria were applied: IQ < 80, sensory handicaps, motor disabilities, chronic diseases, any medication of the child, and insufficient knowledge of the German language in either the parent or the child. Table 1 shows the descriptive data. Examinations were carried out and video recorded at our lab by investigators who were blinded to the child’s ADHD and ODD symptom scores. The children’s task performances were scored from the video recordings by trained research assistants, who were also unaware of the ADHD and ODD symptom scores. Parents and child care teachers filled in questionnaires on the child’s symptoms (see below).

### 2.2. Variables

ADHD and ODD symptoms. Symptoms of ADHD and ODD were assessed using the hyperactivity/inattention subscale and the Conduct Problems Scale of the Strengths and Difficulties Questionnaire (SDQ) [18], completed by the mother and a child care teacher. The SDQ is a widely used questionnaire with good psychometric properties. The hyperactivity/inattention subscale discriminates well between children with and without an ADHD diagnosis [19]. Validated cut-off scores are available which discriminate between the normal, borderline, and clinical range of symptoms [18,19,20].

Impulsivity score. Two tasks on hot IC and one task on behavioral approach tendency were conducted. In the Cookie Delay Task by Kochanska [21], the child is instructed to wait for the ringing of a bell before he or she can retrieve a sweet that is covered by a transparent cup. After a practice trial, six trials followed, with delay intervals between 10 and 40 s. According to Carlson [22], we scored the number of trials with a full waiting period [22]. Validity has been demonstrated by associations of the task with ADHD symptoms [23].

In the Gift Wrap Task by Kochanska [21], the child is instructed to sit behind the experimenter and not to look while the experimenter wraps a present for the child (1 min duration). Then, the experimenter leaves the room for 3 min. The child is instructed to wait and not to peak at or touch the present, which is left on the table and covered by a piece of paper. The child’s peeking behavior is scored (e.g., 0 = does not peak or leave their seat, 1 = peaks or leaves their seat) [22]. The task showed significant correlations with ADHD symptoms [17].

The Stranger-with-Toys Task by Asendorpf [24] is a measure of temperamental inhibition vs. approach tendency. The child sits at a table with one rather boring toy. A stranger enters the room carrying a transparent bag of interesting toys, which she successively unpacks and plays with while not attending to the child. After 3 min, she invites the child to play with her along with the toys and continues to talk kindly to the child for a further 2 min. The latency (sec) until the child’s first spontaneous utterance directed to the stranger is scored. The measure has proven to show good stability (.74 across 2 years) and has shown significant associations with parent ratings of everyday approach behavior [24]. The task has been demonstrated to be associated with ADHD symptoms [25]. In the present study, the inter-rater reliability of the three tasks was tested in 20% of the video recordings and proved to be very good (Cookie-Delay Task: ICC = 0.98; Stranger-with-Toys Task: ICC = 0.99; Gift Wrap Task: Kappa = 0.86).

### 2.3. Analytic Strategy

Descriptive statistics of the variables in Study 1 are reported in Table 1. To discriminate between high and low ADHD and ODD symptoms (ADHD and ODD groups), we used the clinical cut-offs of the SDQ, which have also been suggested for the German population [26]. Children scoring at or above the clinical cut-offs (i.e., >6 for the Hyperactivity Scale and >5 for the Conduct Problems Scale), according to parents or teachers (using an or rule [27]), were classified as showing high symptoms of ADHD and ODD, respectively.

Following Breaux et al. [17], to simplify the scoring procedure, and in order to reach identical levels of measurement, the Cookie Delay and the Stranger-with-Toys tasks were transformed into dichotomous (0 or 1) variables. Each task was scored such that 1 indicated low performance, or high impulsive behavior, and 0 indicated high performance, or low impulsive behavior (Cookie Delay Task: 0 = child did not touch or lift the cup or eat the sweet in any of the trials vs. 1 = child touched or lifted the cup or ate the sweet in at least one trial; Gift Wrap Task: 0 = child did not peak or leave their seat vs. 1 = child peeked or left their seat; Stranger-with-Toys Task: 0 = no spontaneous utterance toward stranger in the first episode (i.e., within 300 s) vs. 1 = spontaneous utterance toward stranger in the first episode (i.e., below 300 s)). To build the impulsivity score, the three variables were added up (see Table 1 for descriptive statistics).

In a preliminary analysis, we calculated the unadjusted correlations of the impulsivity score (and the three single tasks) with the parent- and teacher-reported ADHD and ODD scores. To identify potential confounding variables, we analyzed whether the age and gender of child or parental education level were significantly associated with the ADHD, ODD, or impulsivity scores. In further analyses, we adjusted for those variables which showed significant associations. To test our hypotheses, hierarchical logistic regression models [28] were used. Specifically, the ADHD and ODD cut-offs were used as criterion variables and the impulsivity score as predictor variable. Finally, to identify an optimal cut-point of the impulsivity score which yielded the best balance between sensitivity and specificity in the discrimination between children scoring below vs. above the ADHD and ODD cut-offs, a receiver operating characteristic (ROC) analysis was conducted. We aimed to validate this impulsivity cut-off in Study 2.

## 3. Study 1: Results

The age of the child was not associated with the SDQ Hyperactivity/Inattention score, the SDQ Conduct Problems score, the ADHD or ODD groups, or the impulsivity score (r between −0.01 and 0.14). The gender of the child (r between −0.21 and 0.13, *p* = 0.034) and parental education level (r between −0.26 and 0.06, *p* = 0.012) showed significant associations with some of these variables. We therefore adjusted for the gender of the child and parental education level in the logistic regression analyses.

The three single neuropsychological tasks were significantly associated with most of these scores, indicating that the tasks of hot IC, as well as approach tendency, contributed to the overall association (Table 2a).

The ADHD groups and ODD groups significantly differed in the impulsivity score. The effect sizes were of large magnitudes (ADHD groups: t = 3.18, *p* < 0.01; SMD = 0.91, 95% CI: 0.37–1.44; ODD groups: t = 3.14, *p* < 0.001; SMD = 0.99, 95% CI: 0.34–1.64). Controlling for the gender of the child and parental education level in the logistic hierarchical regression models did not change these results (Table 2b).

The ROC curve analysis resulted in an optimal cut-point of ≥2 for the discrimination of children scoring above vs. below the SDQ cut-offs (ADHD groups and ODD groups). The sensitivity and specificity were 0.56 and 0.61 for the ADHD groups and 0.73 and 0.61 for the ODD groups.

## 4. Study 2: Materials and Methods

### 4.1. Participants and Procedure

A sample of n = 138 four-to-five-year-old (m = 58.2, s = 6.2 months) children (*n* = 85, 59% boys) with elevated ADHD symptoms were recruited from child care centers. At recruitment, parents completed an ADHD screening questionnaire (see below). Children who scored in the upper quartile (i.e., exceeded the lower bound of the 95% CI of the 75th percentile of the reference sample) were considered. Exclusion criteria from the study sample were an IQ below 80, chronic diseases involving brain functions, any continuous pharmacological treatment, and insufficient German language skills of the parent or child. Table 3 shows the descriptive data of the sample.

Examinations were carried out at the child care centers by two trained investigators. The first investigator worked with the child while the second investigator scored the child’s performance in the tasks. The investigators were unaware of the child’s ADHD or ODD symptom scores.

A total of 122 children and their parents took part in the second assessment wave at 8 years (m = 101.3, s = 3.8 months). The 17 children who dropped out from the assessment did not differ from the remainder with respect to parent- or teacher-rated ADHD and ODD symptoms, the age of the child in months (*t*-scores between −1.17 and 1.30), the gender of the child (χ^2^(1) = 0.29), and paternal education level (χ^2^(3) = 0.30). However, children who did not attend the 8 year assessment had mothers with lower education levels (χ^2^(3) = 8.48, *p* = 0.037). The 8 year assessment was carried out at our lab. An investigator (psychologist), who was blind to all data of the first assessment wave, conducted the clinical interview with the mother.

### 4.2. Variables

#### 4.2.1. Assessments at 4–5 Years (Wave 1)

ADHD and ODD symptoms. The parents and child care teachers completed the ADHD rating scale (FBB-ADHS-V) of the Diagnostic System for Psychiatric Disorders (DISYPS-II) [29]. The parent and teacher versions have shown high internal consistency (Cronbach’s α: 0.94 and 0.93) and discriminated well between children with and without ADHD diagnoses. ODD symptoms were measured by the FBB-SSV questionnaire of the DISYPS-II, which also showed good internal consistency (α = 0.87) and discriminated well between children with and without ODD diagnoses [30].

Impulsivity. Impulsivity was measured according to the procedure described above (Study 1). In the present study, inter-rater reliability for the three neuropsychological tasks was checked in 20% of cases and proved to be very good (Stranger-with-Toys: K = 0.87, Cookie-Delay: K = 0.90, Gift-Wrap: K = 0.95).

#### 4.2.2. Assessments at 8 Years (Wave 2)

*ADHD and ODD symptoms and diagnoses*. At the age of 8 years, the ADHD and ODD/CD diagnostic modules of the Child and Adolescent Psychiatric Interview (CAPA) by Angold et al. [31] were conducted with the mother. The CAPA is a well-validated, widely established clinical interview which allows clinical diagnoses to be made according to the DSM-5 [32]. Of the 121 children, *n* = 29 (24%) received an ADHD diagnosis. 

The ADHD and ODD symptoms of the child were additionally assessed by parent and teacher questionnaires. Parents and school teachers completed the FBB-ADHS of the DISYPS-III [33]. The questionnaires captured ADHD symptoms according to ICD-10 and DSM-5 and showed good psychometric properties, as reported above. For the assessment of ODD symptoms, the Conduct Problems Scale of the Strengths and Difficulties Questionnaire (SDQ) [18] was completed by the school teachers and the mothers.

### 4.3. Analytic Strategy

Descriptive statistics of the variables in Study 2 are reported in Table 3. At Wave 2, besides the ADHD diagnoses, dimensional ADHD and ODD symptom scores were created. For this purpose, z-transformed scores of the parent and teacher questionnaire scales and the dimensional CAPA symptom scores were summed up (ADHD score: parent FBB-ADHS, teacher FBB-ADHS, CAPA-ADHD score: r between 0.59, *p* < 0.001 and 0.80, *p* < 0.001; ODD score: parent SDQ–Conduct Problems scores, teacher SDQ–Conduct Problems scores, CAPA-ODD score: r between 0.30, *p* = 0.004 and 0.51, *p* < 0.001).

In a preliminary analysis, the unadjusted correlations of the impulsivity score (and the single tasks) with all ADHD and ODD scores (symptom scores and diagnoses) were calculated. Hierarchical logistic regression models were used to test whether the impulsivity score predicted the 8 year ADHD diagnosis. In the first regression model (Model 1), the gender of the child and parental education level were controlled (i.e., the variables were introduced at Step 1, followed by the impulsivity measure (Step 2)). In a second regression model (Model 2), we additionally controlled for the Wave 1 mother- and teacher-reported ADHD symptom scores (Step 1) in order to test whether the impulsivity score contributed to the prediction of the Wave 2 ADHD diagnosis over and above the concurrent ADHD symptoms. Finally, ROC analysis on the prediction of the Wave 2 ADHD diagnoses was used to validate the impulsivity cut-off score identified in Study 1. Logistic regression analyses on the prediction of ADHD diagnosis (Models 1 and 2) by the impulsivity cut-off score were conducted to explore the predictive effects.

## 5. Study 2: Results

*Preliminary analyses.* The impulsivity score was significantly associated with the Wave 1 and 2 ADHD and ODD scores (Table 4). Children with and without an ADHD diagnosis at Wave 2 differed significantly in the impulsivity measure (t = 3.72, *p* < 0.001; SMD = 0.80, 95% CI: 0.37–1.23).

*Incremental predictive validity of the impulsivity score.* The impulsivity score significantly predicted the ADHD diagnosis over and above the gender of the child and maternal education level (Table 5, Model 1, and Step 2a). Additional adjustment for the Wave 1 mother- and teacher-reported ADHD symptoms (Table 5, Model 2, and Step 2a) resulted in a significant incremental predictive effect. Impulsivity thus predicted an ADHD diagnosis at 8 years over and above concurrent ADHD symptoms, the gender of the child and maternal education level.

ROC analysis on the prediction of ADHD diagnosis by the impulsivity measure resulted in the same optimal cut-point (i.e., ≥2) as in Study 1: sensitivity for the prediction of the 8 years ADHD diagnosis was 0.76, and the specificity was 0.70. The impulsivity cut-off score significantly predicted the Wave 2 ADHD diagnosis (Models 1 and 2). The effect sizes (odds ratios) were of medium to large magnitudes and indicated 6.6-fold (Table 5, Model 1, and Step 2b) and 5-fold (Table 5, Model 2, and Step 2b) increased odds of developing ADHD until school age if preschool impulsivity was high.

## 6. Discussion

High impulsivity has been thought to represent an early developing vulnerability marker for ADHD and externalizing disorders. The construct overlaps with hot IC and comprises deviations of bottom-up and top-down control processes. We aimed at an assessment of the concurrent and predictive validity of a task-based, neuropsychological measure of preschool-age impulsivity. The measure combines tasks for hot IC and approach motivation. In a first cross-sectional study, the impulsivity measure significantly predicted parent- and teacher-reported ADHD and ODD symptoms above a clinical cut-off. The effect sizes were large; thus, concurrent validity was shown. In the second longitudinal study involving preschoolers with significant ADHD symptoms, the impulsivity measure predicted an ADHD diagnosis at 8 years over and above concurrent parent- and teacher-rated ADHD symptoms. Using a cross-validated cut-point, high impulsivity indicated 6.6-fold increased odds of developing ADHD up to school age and 5-fold increased odds over and above the odds associated with prominent parent- and teacher-reported ADHD symptoms, male gender, and a low parental education level. The impulsivity measure thus showed considerable incremental predictive validity.

Longitudinal research on the prediction of ADHD or externalizing disorders by measures of hot IC or trait impulsivity is rare. The few existing studies found that performance in hot IC tasks were significantly associated with ADHD symptoms developing later. Results differed on the question of whether the neuropsychological tasks predicted an increase in ADHD symptoms (i.e., showed incremental predictive effects over and above the concurrent ADHD symptoms). In a high-risk sample, Breaux et al. [17] adjusted for preschool ADHD symptoms but did not find a significant incremental predictive effect of two hot IC tasks. In our longitudinal study (Study 2) involving a comparable high-risk sample, we included a task on behavioral approach tendency, which might have added a significant facet of early trait impulsivity. Moreover, we simplified and evaluated the scoring procedure of the neuropsychological tasks, resulting in very good inter-rater reliability of the tasks. These aspects might have contributed to our results.

We found that a neuropsychological impulsivity measure improved the identification of children at risk of developing ADHD until school age. High impulsivity added 10% of the explained variance in the school-age ADHD diagnoses over and above the preschool ADHD symptoms, the gender of the child, and the parental education level. In future research, it would be worthwhile to further evaluate (e.g., by analyzing the prediction of CD in late childhood) and improve (e.g., by adding further tasks to the battery) the neuropsychological impulsivity measure. Such a measure can be useful from a clinical perspective, enabling children who are actually at risk of developing ADHD to be identified early on and receive support and treatment. False positive decisions could be avoided in those who are likely to outgrow their behavior problems.

The presented research on the validity of our neuropsychological impulsivity summary score has several strengths, such as the use of two independent samples, the use of different, well-validated measures of ADHD and ODD symptoms as criterion variables, and a theory-driven selection of neuropsychological tasks. Limitations, however, might be seen in the rather small sample sizes of the two studies and the inclusion of just three neuropsychological tasks. Our primary intention was to conduct cross-validation of the link between the neuropsychological impulsivity measure and ADHD and ODD. Though the sample sizes of the two studies were rather limited, the result that the measurement adjusted on the basis of the first study showed good predictive validity, and the second study points to the generalizability of the link. As already mentioned, an extension of the impulsivity measure by further neuropsychological tasks on hot IC or approach tendency could increase the reliability and validity of measurements. Furthermore, it could be worthwhile to substitute the Gift Wrap Task with another task, due to the difficulty to conduct the task’s second part outside of a video lab.

## 7. Conclusions

To conclude, in preschool age trait impulsivity can be validly measured by a neuropsychological task-based measure. The measure proved to be useful for ADHD risk prediction. In longitudinal research on developing ADHD and externalizing disorders the impulsivity measure might be useful for an unbiased, valid assessment of an indicator of the early emerging vulnerability.

## Figures and Tables

**Table 1 brainsci-11-00060-t001:** Descriptive statistics for Study 1.

		***n* = 102**
Gender *n* (%)		
Male		47 (46.1)
Female		55 (53.9)
Age in months m(s)		54.6 (11.5)
Education Level of Mother: *n* (%)		
Basic education		11 (10.8)
Work qualification		37 (36.3)
High school		24 (23.5)
College/university		30 (29.4)
Education Level of Father: *n* (%)		
Basic education		19 (18.6)
Work qualification		19 (18.6)
High school		19 (18.6)
College/university		39 (38.2)
(Did not respond)		6 (5.9)
SDQ-Hyperactivity/Inattention Scale		
Parent	m (s)	2.8 (2.4)
Teacher	m (s)	2.9 (2.7)
SDQ Conduct Problems Scale		
Parent	m (s)	2.6 (2.0)
Teacher	m (s)	1.7 (1.8)
Impulsivity score	m (s)	1.7 (1.0)

Note: SDQ = Strengths and Difficulties Questionnaire; impulsivity score = sum of Cookie Delay, Gift Wrap, and Stranger-with-Toys tasks.

**Table 2 brainsci-11-00060-t002:** Study 1: Associations of impulsivity with ADHD and ODD variables. (**a**). Associations of the impulsivity score and the three single tasks with ADHD and ODD symptom scores; (**b**). Associations of the impulsivity score with ADHD and ODD groups.

(**a**)				
	**Impulsivity Score**	**Cookie-Delay Task**	**Gift-Wrap Task**	**Stranger-with-Toys Task**
SDQ Hyperakt/inatten. parent	0.37 ***	0.16	0.26 *	0.34 ***
SDQ Hyperakt/inatten. teacher	0.44 ***	0.21 *	0.34 ***	0.30 **
SDQ Conduct problems parent	0.22 *	0.14	0.24 *	0.09
SDQ Conduct problems teacher	0.35 ***	0.28 **	0.28 **	0.11
(**b**)				
Logistic hierarchical regression model Step 1: gender of child, parental education Step 2: impulsivity score				
	Step 2 Change Statistics			
	OR (95%CI)	χ^2^		*p*
Criterion variable				
ADHD groups ^†^				
≥ clinical cut-off (*n* = 18)	2.48			
< clinical cut-off (*n* = 71)	(1.34–4.60)	9.84		0.002
ODD groups ^‡^				
≥ clinical cut-off (*n* = 11)	2.73			
< clinical cut-off (*n* = 79)	(1.30–5.80)	8.14		0.004

SDQ: Strengths and Difficulties Questionnaire; *** *p* < 0.001; ** *p* < 0.01; * *p* < 0.05. ^†^ SDQ hyperactivity/inattention score ≥ 7 according to parent or teacher, ^‡^ SDQ Conduct problems score ≥ 6 according to parent or teacher; 95%CI: 95% confidence interval; OR: odds ratio.

**Table 3 brainsci-11-00060-t003:** Descriptive statistics of Study 2.

		*n* = 138 Children
Gender *n* (%)		
Male		85 (59.0)
Female		63 (41.0)
Age in months at wave 1 m(s)		58.2 (6.2)
Age in months at wave 2 m(s)		101.3 (3.8)
Education level of mother: *n* (%)		
Basic education		18 (13.0)
Work qualification		57 (41.3)
High school		21 (15.2)
College/university		42 (30.4)
Education level of father: *n* (%)		
Basic education		29 (21.0)
Work qualification		35 (25.4)
High school		29 (21.0)
College/university		41 (29.7)
(Did not respond)		4 (2.9)
Wave 1		
ADHD (FBB-ADHS-V) parent	m (s)	1.3 (0.4)
ADHD (FBB-ADHS-V) teacher	m (s)	0.9 (0.7)
ODD (FBB-SSV) parent	m (s)	0.7 (0.5)
Impulsivity score	m (s)	1.7 (0.8)
Wave 2		
ADHD diagnosis *n* (%)		
Yes		29 (24.0)
No		91 (75.2)
ODD diagnosis *n* (%)		
Yes		7 (5.8)
No		113 (94.2)
ADHD score (CAPA)	m (s)	6.3 (4.0)
ODD score (CAPA)	m (s)	1.8 (1.5)
ADHD (FBB-ADHS) parent	m (s)	1.0 (0.6)
ADHD (FBB-ADHS) teacher	m (s)	0.7 (0.7)
ODD (FBB-SSV) parent	m (s)	0.7 (0.5)
SDQ Conduct problems parent	m (s)	2.2 (2.0)
SDQ Conduct problems teacher	m (s)	3.2 (2.0)

Note: SDQ = Strengths and Difficulties Questionnaire; Impulsivity score = sum of Cookie Delay, Gift Wrap, and Stranger-with-Toys tasks.

**Table 4 brainsci-11-00060-t004:** Concurrent and longitudinal associations between impulsivity and the ADHD and ODD scores in Study 2.

	Impulsivity Score w1
ADHD symptoms parent w1	0.33 ***
ADHD symptoms teacher w1	0.27 **
ODD symptoms parent w1	0.18 *
ADHD diagnosis w2	0.32 ***

*** *p* < 0.001. ** *p* < 0.01. * *p* < 0.05; w = wave.

**Table 5 brainsci-11-00060-t005:** Study 2: Logistic regression analyses on the prediction of 8-years ADHD by impulsivity (score and cut-off).

Prediction of 8-Years ADHD Diagnosis					
		**Step 2 Change Statistics**			**Whole Model**
**Model 1a,b**		**OR (95%CI)**	**χ^2^ (*df*)**	**χ^2^ (*df*)**	***R*^2^_Nagelkerke_**
Step 1	Gender, maternal education				
Step 2a	Impulsivity score w1	2.5 (1.4–4.7)	9.9 (1) *p* = 0.002	19.2 (3) *p* < 0.001	0.22
Step 2b	Impulsivity cut-off w1	6.6 (2.5–17.8)	16.1 (1) *p* = 0.001	25.3 (3) *p* < 0.001	0.28
		**Step 2 Change Statistics**		**Whole Model**	
**Model 2a,b**		**OR (95%CI)**	**χ** **^2^** **(*df*)**	**χ** **^2^** **(*df*)**	***R*** **^2^** **_Nagelkerke_**
Step 1	Gender, maternal education, ADHD parent w1, ADHD teacher w1				
Step 2a	Impulsivity score w1	2.0 (1.1–3.7)	4.5 (1) *p* = 0.027	27.8 (5) *p* < 0.001	0.31
Step 2b	Impulsivity cut-off w1	5.0 (1.8–14.0)	10.0 (1) *p* = 0.002	32.9 (5) *p* < 0.001	0.36

OR = odds ratio; w = wave.

## Data Availability

The data presented in this study are available on request from the corresponding author. The data are not publicly available due to privacy and ethical issues.

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
