# Peer review of "Impulsivity as Early Emerging Vulnerability Factor—Prediction of ADHD by a Preschool Neuropsychological Measure"

_brainsci, 2021, doi:10.3390/brainsci11010060_

Round 1
Reviewer 1 Report
This article investigated the clinical utility of a task-based neuropsychological impulsivity measure for preschool children. Focusing on impulsivity (or hyperactivity) is legitimate since it is difficult to measure inattention for preschoolers. The article is of great clinical value especially in showing predictive validity of the impulsivity measures during preschool years. However, authors should address these flaws listed below.
- When making Impulsivity score, scores of each impulsivity scale were just added up. In other words, each scale was not weighed differently but evenly. Could you show the rationale for this method according to Breaux et al. for readers? Learning from Table 2, I have an impression that a score of Gift-Wrap task should be weighed more heavily than other measures or only the Gift-Wrap task is good enough since the shorter it takes in practicing tasks for preschoolers, the better it gets.
- All measures used in this study including tasks and questionnaires were validated in German?
- As far as I read this article, I couldn’t find the comments about ADHD subtypes. My question is if ADHD inattentive type can be detected as easily as ADHD combined type by the impulsivity measures. Could authors give us any insight about this question?
Minor points
- Statistics-related t, r, and p should be italic.
- In the Table 2, figures in (95% Ci) should be shown to two places of decimals.

Author Response
1. "When making Impulsivity score, scores of each impulsivity scale were just added up. In other words, each scale was not weighed differently but evenly. Could you show the rationale for this method according to Breaux et al. for readers? Learning from Table 2, I have an impression that a score of Gift-Wrap task should be weighed more heavily than other measures or only the Gift-Wrap task is good enough since the shorter it takes in practicing tasks for preschoolers, the better it gets."
This is correct. We used equal weightings of the single scores. It might be that the prediction would have been better if we had calculated weighted scores (e.g. by means of a factor analysis, as we did in previous articles). Here, we decided to use the sum of the unweighted scores because weightings would depend on the specific correlations within a specific sample. At this time, we aimed at the development of a most simple and robust composite score. However, as we mentioned, future improvement of the composite measure is worthwhile.
2. "All measures used in this study including tasks and questionnaires were validated in German?"
Yes, they are.
3. "As far as I read this article, I couldn’t find the comments about ADHD subtypes. My question is if ADHD inattentive type can be detected as easily as ADHD combined type by the impulsivity measures. Could authors give us any insight about this question?"
I agree, that is an important and interesting question. We are just going to analyze preschool- and school-age neurocognitive data in a further article on the age-dependent presentation of hyperactivity/impulsivity symptoms and inattention symptoms. Therefore, we would like to refrain from reporting on these results in the present paper, if possible.
Minor points
- I do not understand, the characters or the numbers or both?
- Thank you, we added the decimal places.
Reviewer 2 Report
I am delighted to read the paper. Thank you for giving me the opportunity to learn about this issue.
I only have a few comments,
If the samples are not the same, the authors must explain why they published these two studies at the same time. At least, they could include a paragraph about the relation between the two studies.
In the second study, wave 1, Symptoms of ADHD and ODD were assessed using the SDQ, and Impulsivity was assessed using the three variables, Cookie delay, the Stranger-with-toys, and Gift-Wrap task. To build the impulsivity score, were added-up but they did not write in table 3.
Were the same researchers working in the two studies? If the researchers were not the same how could they
This study has limitations but it is a very interesting study and difficult to do. Need a lot of time, resource, and researchers
Author Response
"I am delighted to read the paper. Thank you for giving me the opportunity to learn about this issue.
I only have a few comments,"
Thank you for your positive evaluation.
"If the samples are not the same, the authors must explain why they published these two studies at the same time. At least, they could include a paragraph about the relation between the two studies."
The samples are not the same! This is an important issue. As we pointed to in the introduction we wished to validate the task-based impulsivity measue. To show that the measure is associated with ADHD in the two independent samples underscores the validity of the measure. Moreover, we used the cut-off score which we developed in the first sample for the prediction of ADHD in the second longitudinal sample. This is quasi a cross-validation which is required in the scope of a validation of a predictor.
"In the second study, wave 1, Symptoms of ADHD and ODD were assessed using the SDQ, and Impulsivity was assessed using the three variables, Cookie delay, the Stranger-with-toys, and Gift-Wrap task. To build the impulsivity score, were added-up but they did not write in table 3."
Thank you, we now added how the measure was calculated to Table 3
"Were the same researchers working in the two studies? If the researchers were not the same how could they"
With the exception of the authors, not the same researchers. The different research assistants and students were instructed by myself.
"This study has limitations but it is a very interesting study and difficult to do. Need a lot of time, resource, and Researchers"
Yes, that is true and as mentioned to improve the psychometric properties of the impulsivity measure is a issue of future research.
Reviewer 3 Report
This is a very well-written and interesting study that assesses the concurrent and predictive value of measures of pre-school impulsivity for ADHD and ODD symptoms. Especially, the results of the longitudinal study are very interesting and of great clinical utility to health professionals in the field of ADHD and psychologists.
Minor point:
line 64: replace 'has' with 'have'.
Author Response
"This is a very well-written and interesting study that assesses the concurrent and predictive value of measures of pre-school impulsivity for ADHD and ODD symptoms. Especially, the results of the longitudinal study are very interesting and of great clinical utility to health professionals in the field of ADHD and psychologists.
Minor point:
line 64: replace 'has' with 'have'"
Thank you for the positive Evaluation which we very much apprechiate. We changed "has" to "have" in line 64.
Thank you for finding the mistake!